# Effects of Graphene Nanosheets with Different Lateral Sizes as Conductive Additives on the Electrochemical Performance of LiNi_0.5_Co_0.2_Mn_0.3_O_2_ Cathode Materials for Li Ion Batteries

**DOI:** 10.3390/polym12051162

**Published:** 2020-05-19

**Authors:** Ting-Hao Hsu, Wei-Ren Liu

**Affiliations:** Department of Chemical Engineering, R&D Center for Membrane Technology, Research Center for Circular Economy, Chung Yuan Christian University, 200 Chung Pei Road, Chung Li District, Taoyuan City 32023, Taiwan; davy855255@gmail.com

**Keywords:** cathode, graphene, Li-ion batteries, conductive additives

## Abstract

In this study, we focus on lateral size effects of graphene nanosheets as conductive additives for LiNi_0.5_Co_0.2_Mn_0.3_O_2_ (NCM) cathode materials for Li-ion batteries. We used two different lateral sizes of graphene, 13 (GN-13) and 28 µm (GN-28). It can be found that the larger sheet sizes of graphene nanosheets give a poorer rate capability. The electrochemical measurements indicate that GN-13 delivers an average capacity of 189.8 mAh/g at 0.1 C and 114.2 mAh/g at 2 C and GN-28 exhibits an average capacity of 179.4 mAh/g at 0.1 C and only 6 mAh/g at 2 C. Moreover, according to the results of alternating current (AC) impedance, it can be found that the GN-28 sample has much higher resistance than that of GN-13. The reason might be attributed to that GN-28 has a longer diffusion distance of ion transfer and the mismatch of particle size between NCM and GN-28. The corresponding characterization might provide important reference for Li-ion battery applications.

## 1. Introduction

The scientific and technological understanding of batteries has been developed rapidly in recent years because of the needs of the electronic devices market. Li-ion batteries are major energy storage devices that are used in many different electronic devices, such as cellphones, watches, computers, etc. [1]. The requirements of the electric vehicle market have also been extended in recent years. To meet the electric vehicle market’s requirements, energy density, operating voltage and stability play fundamental roles in the battery. LiFePO_4_ [2] is one of the most widely used cathode materials in electric vehicles and it was intensively studied for many years, but there are some problems that should be solved, such as lower energy density, lower operating voltage and lower capacity [3,4]. In order to get high energy density, the layer structures of LiCo_x_Ni_y_Mn_z_O_2_ (x + y + z = 1) are widely studied because of their high operating voltage and higher specific capacity. Compared to LiFePO_4_, LiCo_x_Ni_y_Mn_z_O_2_, namely NCM-based cathodes, demonstrated much better electrochemical performance, such as higher specific capacity, operating voltage and energy density [5,6,7,8,9,10,11,12,13,14]. Thus, many efforts have been made to improve the electrochemical performance of NCM-based cathodes from the viewpoints of electronic conductivity and structure stability. Changing the chemical composition and surface morphology modifications of NCM-based cathode materials are efficient methods to enhance their electronic conductivity or electrode stability, such as oxide coatings [15,16,17,18,19] and doping techniques [19,20,21,22]. Zhu et al. [15] proposed a LiNi_0.5_Co_0.2_Mn_0.3_O_2_/LiFePO_4_ core-shell structure. By LiFePO_4_ coating, the electrochemical performance and safety of the composite are greatly improved by pouch fill cells tests. Yang et al. reported a one-dimensional Nb-doped LiNi_1/3_Co_1/3_Mn_1/3_O_2_ cathode nanostructure. By Nb doping, the reversible capacity, and structure stability as well as the electronic conductivity and cycle performance of the LiNi_1/3_Co_1/3_Mn_1/3_O_2_ cathode could be enhanced. The composition-optimized Nb-doped LiNi_1/3_Co_1/3_Mn_1/3_O_2_ cathode achieved a high discharge capacity of 200.4 mAh·g^−1^ at 0.1 C. Even at 5 C, the modified cathode still had a high capacity of 118.7 mAh·g^−1^ after 200 cycles [20]. As well as oxide coating and element doping modifications, adding electronic conductive additive [23,24,25] is also an efficient way to enhance the electronic conductivity of cathode materials and cycle performance. Carbon materials [26], of course, are the most commonly used as conductive additives because of their high conductivity and chemical stability. For instance, Du et al. [24] reported carbon nanotubes (CNTs) and Super-P as the conductive additive to make the slurry of lithium nickel manganese cobalt oxide (NMC) that can build the three-dimensional structure network to improve its electronic conductivity. The content of active materials increased from 90 to 96 wt %. After 200 cycles the retention in full pouch cell was still 99.4%, which was better than the carbon black. Nowadays, much attention has been drawn to graphene-based materials as conductive additives in cathode materials for Li-ion batteries. Undoubtedly, graphene nanosheets (GN) [11,12,27,28,29,30,31,32,33,34,35,36,37,38,39,40,41,42,43] are potential candidates because of their unique chemical properties, high mechanical strength, chemical tolerance and excellent conductivity. Wang et al. [42] found that by only 3 wt % graphene loading, the specific capacity of LiFePO_4_ cathode could be dramatically enhanced from 150 to 178 mAh·g^−1^. For LiCoO_2_ and Li(Ni_1/3_Mn_1/3_Co_1/3_)O_2_ cathodes, the maximum specific capacity of 156 and 168 mAh·g^−1^ were also enhanced by using graphene as a conductive additive. Wang’s group further demonstrated that by adding liquid-exfoliated graphene, the electrode kinetic energy and reversibility of cathodes were greatly enhanced. In addition, graphene-based additives in cathode materials could also shorten Li-ion diffusion paths and reduce polarization in cathode material particles. The liquid-exfoliated graphene used as the conductive additive in cathode materials decreased the content of conductive additive without reducing the electrode conductivity [44]. The results indicated that cathodes with multi-carbon additives displayed better improvement in capacity and rate capability. Difference kinds of carbon additives were found to have great effects on the improvement of electrochemical performance. CNTs and graphene make up the shortage of carbon black. When all the carbon additives were used together, NCM particles were well covered by graphene, with carbon black particles coating onto the surface, and the whole sheet was connected by CNTs, offering electrons both “short path” and “long path” highways to transport [45]. The lateral sizes of graphene nanosheets are also a key issue for the applications of graphene additives in cathode materials. In Liu’s study, the different graphene size affected the electrochemical performance of LiFePO_4_ cathode materials. The results indicated that the smallest size of graphene had better electrochemical performance (165 mAh·g^−1^). The specific capacity and rate performance of LiFePO_4_ electrodes tend to worsen with increases of the size of graphene, because the length of ionic transport path got longer [46].

In this study, we studied the effects of different graphene nanosheets sizes on LiNi_0.5_Co_0.2_Mn_0.3_O_2_ cathode materials. Graphene and carbon black form three-dimensional structure networks by point-to-plane contact with the graphene and carbon black. The electrochemical measurements indicate that GN-13 delivers an average capacity of 189.8 mAh/g at 0.1 C and 114.2 mAh/g at 2 C and GN-28 exhibits an average capacity of 179.4 mAh/g at 0.1 C and only 6 mAh/g at 2 C, respectively. 

## 2. Experimental

### 2.1. Electrode Composition and Characterizations

The composition of electrodes consisted of LiNi_0.5_Co_0.2_Mn_0.3_O_2_ (NCM from MIT Advanced Material Co., Ltd, New Taipei City, Taiwan; D_50_ = 15 μm) as the active material, polyvinylidene fluoride (PVDF) as binder and graphene ((graphene (GN, Hengwang^®^, our cooperated company, Xuancheng city, China) and carbon black (Super-P, Timcal^®^, Shanghai City, China) as conductive additives in the weight ratio 91:4:3:2. Commercial graphene nanosheets (5–10 layers) with different sizes were used in this study. For comparison, two different lateral sizes of graphene nanosheets (denoted as GN-13 and GN-28) were synthesized by using the cavitation process via fixing the chamber pressure at 2000 bar and changing the cycling times to 3 times or 12 times to obtain GN-13 (smaller lateral size) and GN-28 (bigger lateral size), respectively [47]. The NCM powders were investigated by powder X-ray diffraction (XRD, Bruker D8 Advance Eco) with Cu Kα radiation (λ = 1.5418 Å). Scanning electron microscopy (SEM, Hitachi S-4100 Tokyo, Japan) and energy dispersive X-ray spectroscopy (EDS) analyzed the morphology and structure of powders. The lateral size was analyzed by dynamic light scattering (DLS, Malvern Zetasizer Nano ZS90, Malvern, United Kingdom). High resolution transmission electron microscopy (HRTEM, JEOL JEM-2100, Guangzhou City, China) images were taken on acceleration voltage of 200 kV. 

### 2.2. Slurry Preparation

The slurry was fabricated as follows: first, polyvinyl difluoride (PVDF, 0.04 g) was added into the graphene (4.6 wt %) suspension in a 10 mL sample vial, stirring smoothly for 2 h by hotplate to make sure the PVDF was fully dissolved. Second, mixing of Super-P (0.02 g) and active material (0.91 g) by powder mixing machine. The Super-P electrode was made in the same method by 5 wt % without adding graphene. Finally, the above mixture was added into N-methyl pyrrolidone (NMP) solution and stirred overnight. The slurry was coated onto aluminum foil (20 μm) by a coating machine and dried in an oven at 80 °C for 30 minutes. The electrodes were punched (Diameter = 14 mm) and dried at 120 °C in a vacuum system overnight to remove the residual water. Electrochemical performance was analyzed by coin cells (CR2032, Xinet International Co. Ltd., Taichung City, Taiwan). 

### 2.3. Electrochemical Measurements

The batteries were assembled in an Ar-gas filled glove box with H_2_O and O_2_ content < 0.5 ppm, using a lithium disk as the counter electrode. The electrolyte consisted of 1 M LiPF_6_ in ethylene carbonate (EC), ethyl methyl carbonate (EMC) and dimethyl carbonate (DMC) (1:1:1 by wt %), with 1 wt % vinylene carbonate (VC). The separators were purchased from Celgard 2325^®^. The electrochemical performance tests for rate performance (0.1, 0.2, 0.5, 1, 2, 5 and 10 C) were analysed by an AcuTech System in the voltage window of 3.0 and 4.5 V at room temperature. The electrodes were analyzed by cyclic voltammograms (CV) (CH Instruments Analyzer CHI 6273E) in the voltage window of 3.0 to 4.5 V with a scan rate of 0.1~0.5 mV·s^−1^. Electrochemical impedance spectroscopy (EIS) measurements were measured on Bio-Logic Science Instruments (VSP-300) with a perturbation amplitude of 5 mV at the frequency range between 100,000 Hz and 0.01 Hz.

## 3. Results and Discussion

Figure 1a displays the X-ray diffraction pattern of the NCM sample. The diffraction peaks of 18°, 36°, 38°, 44°, 48°, 58°, 64°, 65° and 68° could be indexed as the (003), (101), (10-2), (104), (10-5), (009), (10-8), (2-10) and (2-13) planes, respectively, by comparing the standard XRD pattern of NCM peaks with ICSD-242139. The result indicates that the active material was pure-phased with highly crystallinity. Figure 1b shows the SEM image of the surface morphology of NCM. From the SEM image we could observe that the particle size distribution of NCM was from 10 to 15 μm with a sphere-like morphology. Furthermore, from the same results as the SEM observations, the particle size distribution of the NCM was similar and ranged from 8 to 20 μm with wide lateral size of 15 μm (D_50_) (Figure 1c). Figure 1d shows the atomic ratio of the NCM cathode powders and without other signals of elements according to EDS analysis.

Figure 2 shows the SEM images and HR-TEM images of the surface morphology of GN-13 and GN-28, respectively. Figure 2a,b shows the SEM images of GN-13 and GN-28. The nature of few-layered graphene can be seen without significant stacking. Figure 2c,d shows the TEM images of GN-13 and GN-28. The contour of graphene can be clearly seen from the figure, and the graphene nanosheet was only stacked to a few layers, it can be observed to be translucent under the TEM. The edges are wrinkled, which was one of the typical features of graphene, and the small columnar may be dispersant.

Figure 3 shows the AFM (Atomic force microscopy) images and the distribution of thickness for GN-13 and GN-28 samples. We took 30 samples to obtain the information of the average thickness and distribution of thickness. All of the graphene nanosheets were of a thickness approximate to 3 to 5 nm. The few-layered graphene GN-13 and GN-28 had six to 10 layers that prove graphene sheets were of high quality. Figure 4 shows the DLS images of different graphene nanosheets. The average sizes (D_50_) of GN-13 and GN-28 were 13 and 28 μm, respectively. The particle sizes (D_10_) and (D_90_) of GN-13 were 5 and 28 μm and of GN-28 were 11 and 79 μm, respectively.

Figure 5 displays X-ray diffraction patterns of different lateral sizes of graphene nanosheets. The grain size of GN-13 and GN-28 were determined to be 135.516 and 86.277 Å, respectively, by the Scherrer equation:
D = κλ/β/cosθ(1)
where D is the mean size of the crystalline domains, κ is the dimensionless shape factor, λ is the X-ray wavelength (1.5418 Å), β is the line broadening at half the maximum intensity (FWHM) and θ is the Bragg angle.

Figure 6a shows the rate performances of the NCM with different sheet sizes of graphene and Super-P. The GN-13 electrode demonstrates charge capacity with an increased current rate of 189.8, 165.6, 142.2 and 114.2 mAh/g when the current rate increased from 0.1, 0.5, 1 and 2 C, respectively. The GN-28 electrode exhibits charge capacities with increased current rates of 179.4, 134.8, 75.2 and 6 mAh/g at current rate from 0.1, 0.5, 1 and 2 C, respectively. For comparison, we also introduced 5 wt % Super P as a conductive additive in the NCM electrode for the control measurement. The Super-P electrode shows reversible capacity of 168.4, 151.4, 136.6 and 37.2 mAh/g at 0.1, 0.5, 1 and 2 C, respectively. Compare to GN-13/Super-P and GN-28/Super-P, we found that bi-conductive additives gave much better rate capability than Super P. For example, at 2 C, the capacity of GN-13 and Super P were 114.2 and 37.2 mAh/g. The enhancement could be 3.06 times by intruding GN-13/Super P as composite conductive additives. The reason for the dramatically enhancement might be explained that GN-13/Super P provides a shorter point-to-plane structure for the Li-ion transfer path. The structure of the Super-P electrode is the only point-to-point structure in which the C-rate performance was worse at high rate. The results show that the high rate performance of the electrode becomes worse without the addition of graphene. When the current rate is redirected back to 0.1 C, the charge capacity of GN-13 and GN-28 electrodes are recovered to 178.8 and 160.4 mAh/g, respectively. Figure 6b,c are the charge/discharge curves of GN-13 and GN-28, respectively. The charge/discharge electrical potential difference of GN-13 at 90 mAh/g (0.5 C) is 0.219 V. The difference of electrical potential for GN-28 at 90 mAh/g (0.5 C) was 0.754 V. The lower potential difference needs less energy than high potential difference for charge/discharge. It can be inferred from these graphs that GN-28 suffers from severe polarization which results in poor electrochemical conductivity. When the graphene sheet size is approximate with NCM particle sizes this can improve the electrochemical performance and decrease the diffusion distance of ion transfer. Considering each specific capacity at 0.1 C as 100%, the rate capability of GN-13 and GN-28 were investigated by increasing current density from 0.1 to 10 C. Figure 6d shows the capacity retention of different C-rates and the retention of GN-13 at 0.5, 1 and 2 C was about 96%, 87% and 75%, respectively. The capacity retention of GN-28 was about 75%, 42% and 3% at 0.5, 1 and 2 C, respectively.

Figure 7a shows EIS analyses of GN-13 and GN-28 electrodes. The inset shows the equivalent circuits and fitting results after three cycles (0.1 C) at charged state. As shown in Figure 7a, the diameter of the semicircle for the GN-13 electrode was much lower than that of the GN-28 electrode. Semicircles and a straight sloping line from the high to low frequency region are the typical EIS. The internal resistance of the coin cell systems includes the combined resistance from the electrolyte, solid electrolyte interphase (SEI), charge transfer and Li-ion diffusion into the electrode. Thus, we use R_S_, R_f_ and R_CT_ items in the equivalent circuit. R_S_ is the electrolyte resistance at the highest frequency, Rf is the resistance of the SEI film and R_CT_ is the charge transfer resistance at the middle frequency region. The sloping line represents the diffusion of Li-ion at the lower frequency. Depending on the EIS plot, the R_S_, R_f_ and R_CT_ values of GN-13 are 4.14, 123 and 40 Ω, respectively. The R_S_, R_f_ and R_CT_ values of GN-28 are 6.13, 142 and 202 Ω, respectively. R_CT_ decreases when smaller graphene is used to be the conductive agent, owing to shorter diffusion length of ionic transfer in the NCM structure. The ionic conductivity of the GN-13 electrode is higher than that of the GN-28 electrode, which indicates that the R_CT_ of GN-13 electrode is smaller than that of GN-28 electrode. The lower slope of EIS analyses have better ionic diffusivity and conductivity. The diffusion length of ionic transfer in the GN-13 based electrode was shorter than that of the GN-28 based electrode. The following equation can calculate the Li-ion diffusion coefficient (D):
D = R^2^ T^2^/2A^2^ n^4^ F^4^ C^2^ σ^2^(2)
The ideal gas constant is R (8.314 J/K mol), the room temperature is T, the electrode surface area is A (~1.54 cm^2^), the electrons number involved in the reaction is n (n = 0.75), Faraday’s constant is F (96500 C/mol), and the Li-ion concentration is C (0.001 mol/cm^3^). Figure 7b shows the relationship lines between Z’ vs. ω^−^^1/2^ at low frequency region of GN-13 and GN-28. We calculated the diffusion coefficients of Li-ion in GN-13 and GN-28 to be 4.437 × 10^−10^ and 4.227 × 10^−10^ cm^2^/s by the equation, respectively.

Figure 8a shows the cyclic voltammogram (CV) profiles of GN-13 and GN-28 at a scan rate of 0.1–0.5 mV/s in the voltage window of 3–4.5 V versus Li/Li^+^. The curves of both graphene show a similar charge/discharge plateau with the work voltage at 3.0–4.5 V. The diffusion coefficient of lithium-ion in GN-13 and GN-28 can be calculated by the following Randles–Sevcik equation via CV analysis:i_p_/m = 0.4463 (F^3^/RT)^1/2^ n^3/2^ AD^1/2^ Cv^1/2^(3)
The peak current is i_p_ (A), the mass of the active cathode material is m, Faraday’s constant is F (96500 C/mol), the ideal gas constant is E (8.314 J/K mol), the room temperature is T, the number of electrons involved in the reaction is n (n = 0.75), the electrode surface area is A (~1.54 cm^2^), and the concentration of Li-ion is C (0.001 mol/cm^3^). Figure 8b shows that the diffusion coefficient of GN-13 and GN-28 calculated by the slope of cyclic voltammograms. According to the equation, diffusion coefficients of lithium ions in GN-13 and GN-28 are calculated to be 3.23 × 10^−8^ and 2.61 × 10^−8^ cm^2^/s for anodic and 2.16 × 10^−8^ and 1.8 × 10^−8^ cm^2^/s for cathodic reaction, respectively. As shown in Figure 8b, the slope of GN-13 was higher than that of GN-28. The ionic diffusivity of the GN-13 electrode was better than that of the GN-28 electrode. According to Table 1, the separation peak increases from GN-13 to GN-28, which means a lateral size of graphene similar with NCM poses the least polarization. In order observe the distribution of NCM, GN-13 and GN-28 in electrode, Figure 9 shows SEM images of fresh electrodes (without charge and discharge) of the GN-13 electrode (Figure 9a,b) and the GN-28 electrode (Figure 9c,d). It is a little difficult to identify which one is cathode material (NCM) and which one is conductive additives (GN-13, GN-28 or Super P). Thus, we try to label these materials by different colors. The red lines are NCM, the blue and green lines are GN-13 and GN-28, respectively. As shown in Figure 9, GN-28 was too big to cover NCM cathodes. Nevertheless, the size of GN-13 and the NCM cathode was similar. Thus, the rate capability of the NCM cathode might be improved by intruding GN-13 instead of GN-28. Figure 10 is the schematic diagram of lithium ion and electron transport paths in different systems using graphene with different sizes and Super-P as conductive additives. The graphene sheet size is similar to that of NCM in that the distance of electron transport is shorter.

## 4. Conclusions

In summary, we used graphene nanosheets with different lateral sizes as conductive additives in NCM to study the size effects on their electrochemical properties. An effective electronic conducting network can be constructed and significantly improved the electrochemical performance of NCM by graphene nanosheets and Super P as conductive additives simultaneously. With increased graphene sheet sizes, the specific capacity and rate performance tends to get worse for NCM electrodes due to that the distance of the ion transport path is protracted. Size matching is the critical issue for choosing suitable graphene nanosheets as conductive additives for NCM cathodes for Li-ion battery application. Similar sizing matching of GN and NCM could keep effective balance between fast lithium ion diffusion and increased electron transport.

## Figures and Tables

**Figure 1 polymers-12-01162-f001:**
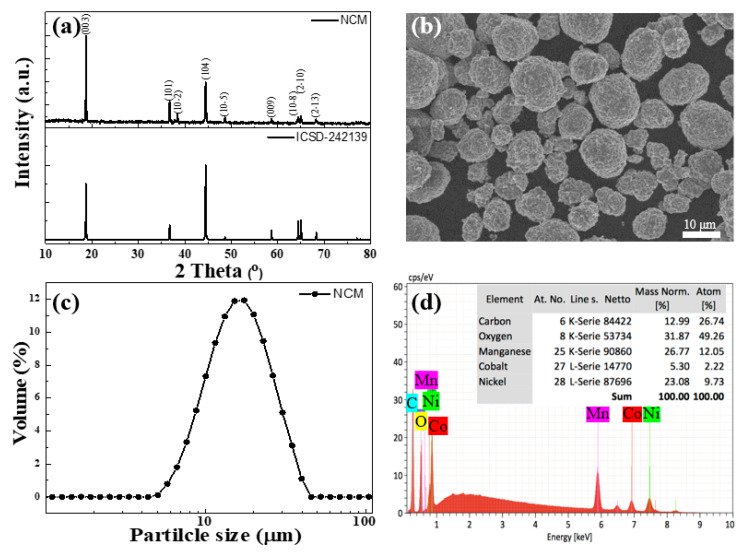
(**a**) X-ray diffraction (XRD) pattern of LiNi0.5Co0.2Mn0.3O2 (NCM), (**b**) scanning electron microscopy (SEM) image of NCM, (**c**) particle size distribution of NCM and (**d**) energy dispersive X-ray spectroscopy (EDS) image of NCM.

**Figure 2 polymers-12-01162-f002:**
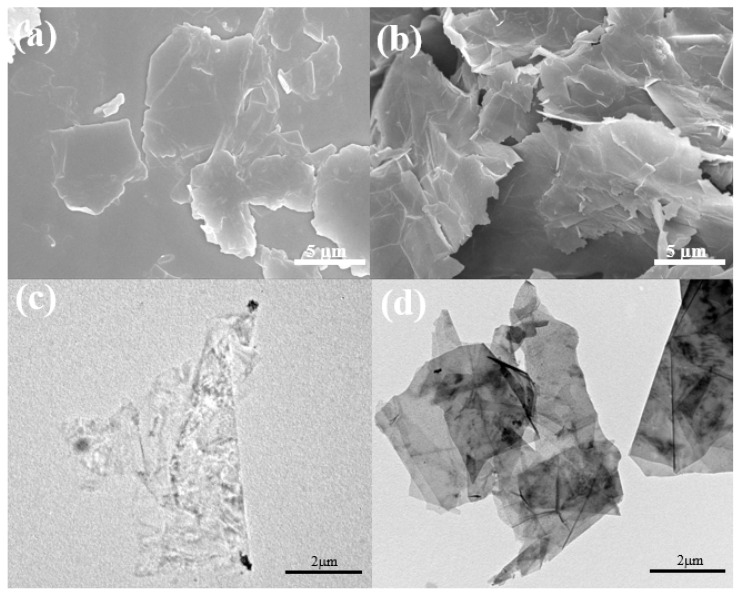
SEM images of different graphene sheet sizes: (**a**) GN-13 and (**b**) GN-28, and transmission electron microscopy (TEM) images of different graphene sheet sizes: (**c**) GN-13 and (**d**) GN-28.

**Figure 3 polymers-12-01162-f003:**
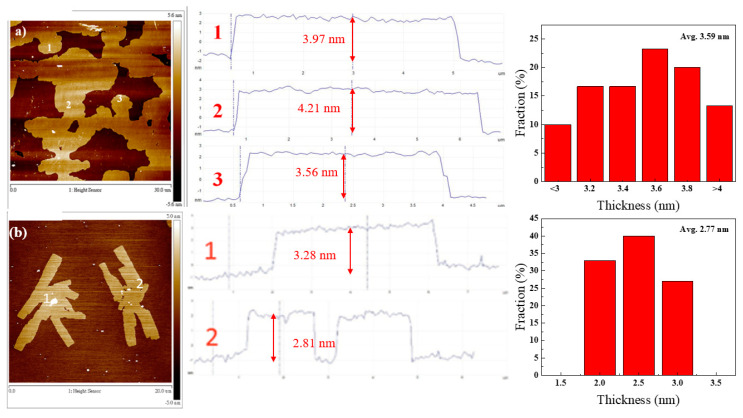
AFM images of different graphene sheet sizes and distribution of thickness of (**a**) GN-13 and (**b**) GN-28 with 30 samples.

**Figure 4 polymers-12-01162-f004:**
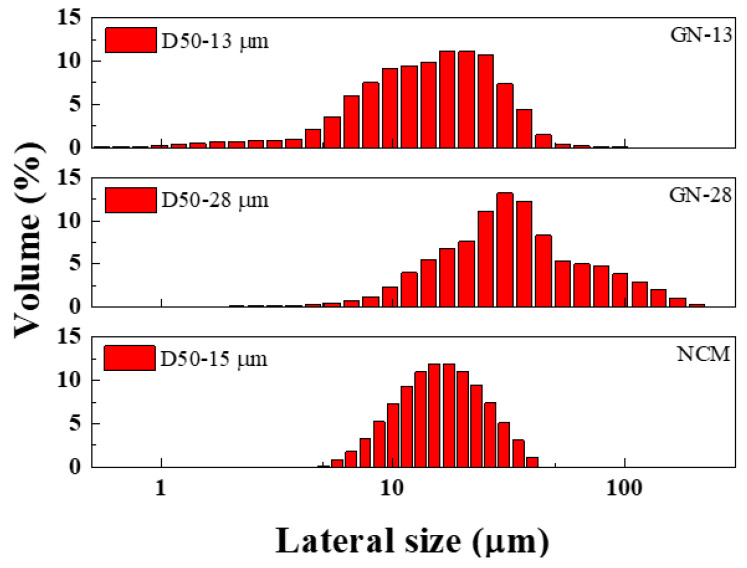
Lateral size distribution of GN-13 and GN-28 and NCM samples.

**Figure 5 polymers-12-01162-f005:**
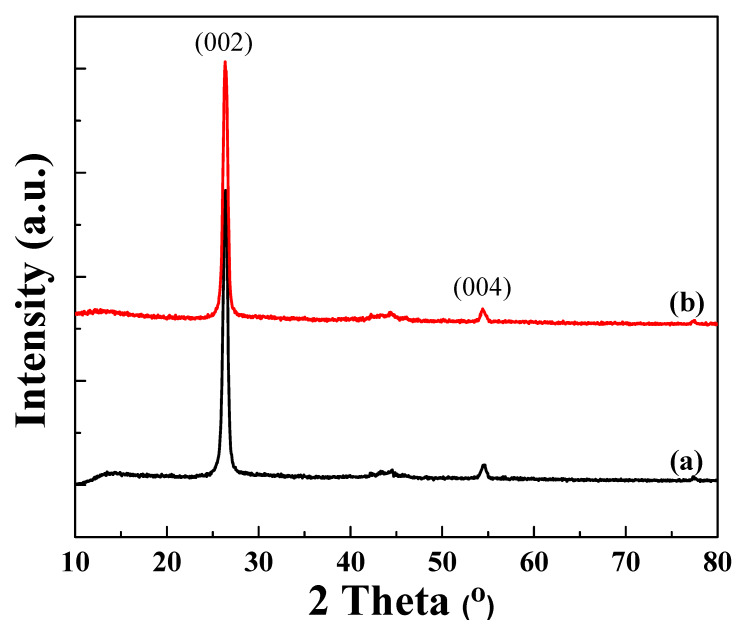
XRD patterns of (**a**) GN-13 (**b**) GN-28 samples.

**Figure 6 polymers-12-01162-f006:**
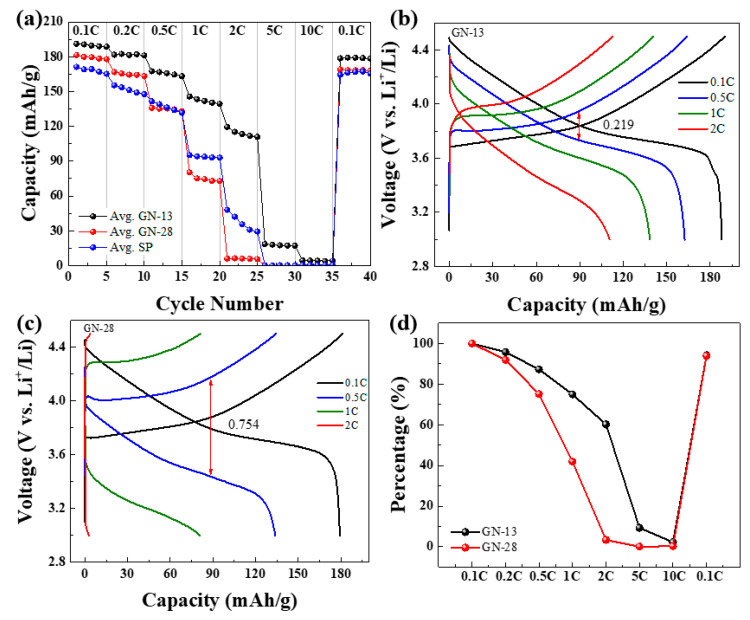
Electrochemical properties of NCM: (**a**) C-rate tests, charge/discharge curves of NCM at different C-rates of (**b**) GN-13 and (**c**) GN-28. (**d**) Retentions of different C-rates.

**Figure 7 polymers-12-01162-f007:**
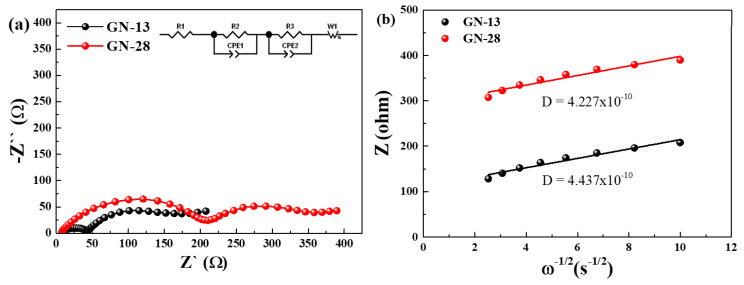
(**a**) Electrochemical impedance spectra (EIS) (Nyquist plots), inset: EIS fit by the equivalent circuit and (**b**) the relationship lines between Z’ vs. ω^−1/2^ at low frequency.

**Figure 8 polymers-12-01162-f008:**
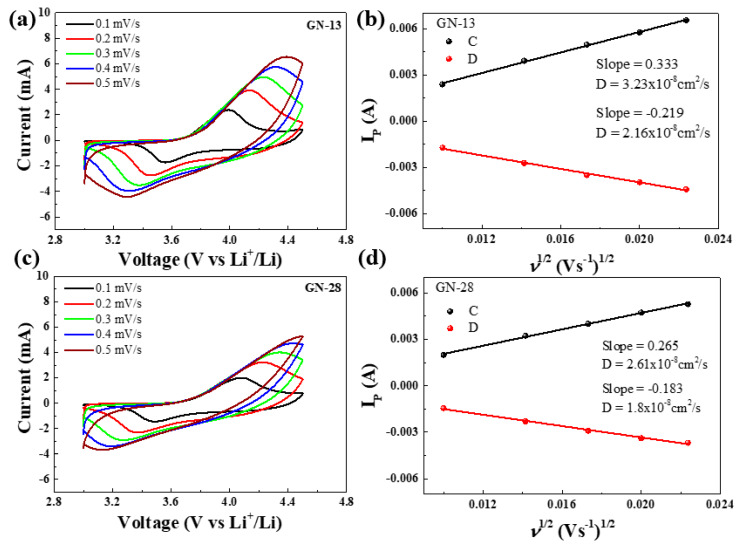
Electrochemical performance of NCM: (**a**) cyclic voltammograms (CV) test of GN-13;(**b**) the relationship lines between I_P_ vs. *ν*^1/2^ of GN-13; (**c**) CV test of GN-28; (**d**) the relationship lines between I_P_ vs. *ν*^1/2^ of GN-28.

**Figure 9 polymers-12-01162-f009:**
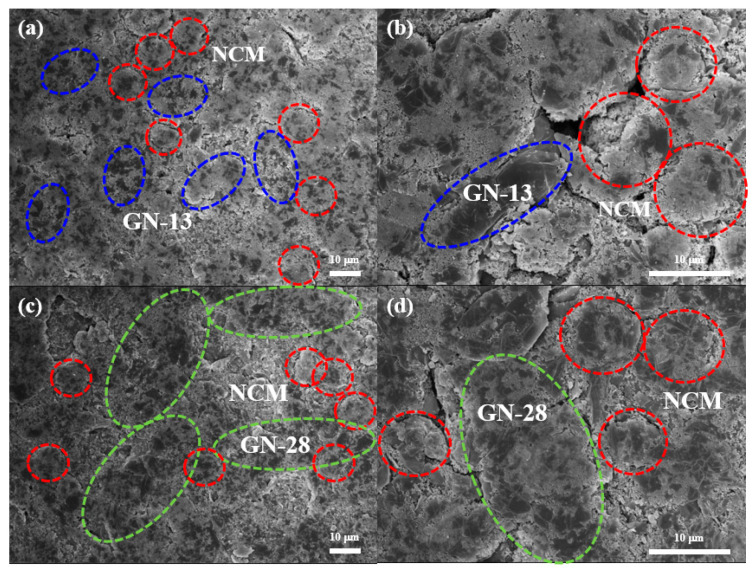
SEM images of fresh electrodes by using different graphene as conductive additives: (**a**) and (**b**) GN-13 electrode; (**c**,**d**) GN-28 electrode. The red dash-line is NCM, the blue dash-line is GN-13 and the green dash-line is GN-28.

**Figure 10 polymers-12-01162-f010:**
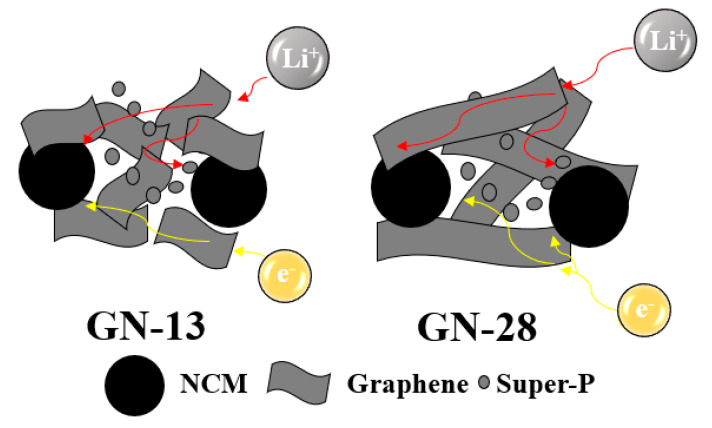
The schematic diagram of lithium ion and electron transport paths in different systems utilizing graphene nanosheets (GN) with different sizes and Super-P as conductive additives.

**Table 1 polymers-12-01162-t001:** Peak potentials obtained from the CV data of GN-13 and GN-28 at 0.1 and 0.5 mV/s.

Samples (Scan Rate)	Anodic Peak (V)	Cathodic Peak (V)	Peak Separation (V)
GN-13 (0.1 mV/s)	3.999	3.555	0.444
GN-28 (0.1 mV/s)	4.079	3.485	0.594

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
