# Peer review of "Effects of Graphene Nanosheets with Different Lateral Sizes as Conductive Additives on the Electrochemical Performance of LiNi0.5Co0.2Mn0.3O2 Cathode Materials for Li Ion Batteries"

_polymers, 2020, doi:10.3390/polym12051162_

Round 1

Reviewer 1 Report

  1. The mechanism about the connection between graphene nanosheets and LiNi5Co0.2Mn0.3O2 is not clear. And whether there are changes in the layered structure before and after the cycling tests need to be confirmed.
  2. Abbreviations should be written in full when they first appear, such as "AC, PVDF " in "abstract and slurry preparations”
  3. There is a big decrement of capacity (capacity fade) for GN-28 at 2C. The provided reason is not satisfactory and the author should explain why such a huge capacity fade happened for this system? If possible, please report the long cycling of LiNi5Co0.2Mn0.3O2 may 100 or 200 cycles.
  4. The author should correlate the electrochemical and cycling properties of the present GN/LMNCO with earlier reports and provide a comparison table.
  5. I noticed plenty of missing verbs and typographical errors, the authors should concern about these errors throughout the manuscript.

Author Response

Thank you.

Reviewer 2 Report

The authors of this paper studied the effect of graphene lateral size on NMC cathode materials, and found that larger sheet size give poorer rate capability. This difference is attributed to longer diffusion length and higher resistance. Detailed characterization was carried out, including EIS and CV measurements. Overall a great piece of carefully studied work. Minor revision is suggested prior to acceptance.

In Fig 1a, the relative intensities of the XRD peaks at about 18 and 44 deg are different from the standard PDF pattern. Please explain.

In Fig 1c, x axis label "particle" is spelt wrongly.

On page 9, the authors wrote that: "diffusion coefficients of Li-ion in GN-13 and GN-28 are calculated to be 4.227x10-10 cm2/s and 4.437x10-10 cm2/s, respectively." The values should be switched. Please check again carefully.

In Fig 8, the x-axis for square root of scan speed should be the greek italic symbol nu (ν) and not the capital letter V.

Graphene has also been applied successfully in many other battery systems but the papers are missing in the reference list. Some classical examples are as following: Chem. Sci. 2014, 5, 1396-1400, Nano Lett. 2011, 11, 7, 2644-2647, ACS Nano 2020, 14, 1, 21-25.

Some relevant battery papers should be cited in the introduction: ACS Nano 2020, 14, 1148-1157, Nano Lett. 2020, 20, 546-552, Energy Storage Mater. 2019, 23, 261-268.

Author Response

Thank you.

Reviewer 3 Report

The authors report a study comparing ~ 13µm laterally long graphene sheets vs. 28µm sheets as conductive additives to NCM/Li batteries, concluding that the shorter graphene sheets and smaller diffusion domains of electrons and Li promote superior electrochemistry as evidenced by prominent capacity differences at ca. 2C rates.

However, the major issue which prevents publication of this work at this time is that graphene is not the sole conductive additive: carbon black is added to the electrode and crucially the amounts are absent/not able to be ascertained from the manuscript's Experimental details. To paraphrase, the authors report that NCM as active material, PVDF binder, graphene, and carbonblack (four components) are added in the weight ratio "91:4:5".

Carbon black is widely used precisely because of its small domains, and so unless the addition of this graphene is shown to improve upon carbon black, it is likely this is a manuscript about how to make worse conductive carbon. Alternately stated- the authors do not have a "0 graphene" control measurement: what happens if all the added carbon was super P?

I suspect the additional Super P prevents the authors from making more substantial claims than the data allows them. The measured Li diffusion coefficients by impedance following 3 cycles is 4.227*10-10 vs. 4.437*10-10 cm2/s between the two materials and this is interpreted, to paraphrase p8L196 and below, as higher ionic conductivity and better ionic diffusivity. Such an increase of at most 5% (which I would lobby needs to be shown to be significant via a T test) I find improbable to explain what turns out to be a capacity difference at 2C of two orders of magnitude. However, I would imagine the diffusion coefficients could be more drastically different if Super P were absent.

The authors should provide SEM images of the active material mixed with super P and both types of graphene. The current SEM images all have different scales and so it is difficult to envision how each of these samples are connected. If the NCM is truly much smaller than G28 and closer in size matching to G13 it should be shown. Notably, in other work (e.g., https://pubs.acs.org/doi/abs/10.1021/acsami.7b12307) it is known that the atomic interface of the active material can more than make up for mismatch on size between carbon and active material.

While the graphene characterization seemed well done, the source is cited as "commercial graphene" with no other details, so it is unknown whether the samples were provided from the same vendor or not. I would be interested in elemental analysis of these graphenes, especially to see if any impurities exist from commercial synthesis. It is well documented, e.g. by Pumera et. al., that metal impurities can drastically change the electrochemistry effects of graphene and composites, (this recent article is a rather humorous reminder https://pubs.acs.org/doi/10.1021/acsnano.9b00184). The possibility that 13 micron graphene may have different impurities than 28 micron graphene is not fully eliminated from the authors characterization in this manuscript.

Author Response

Thank you.

Round 2

Reviewer 3 Report

Regarding the authors response:

  1. The authors satisfactorily addressed the electrode preparation procedure.
  2. The data procured by the authors appears to show their graphene indeed improving the electrode relative to super P carbon. However, such a Figure is provided only for my review and is not included or discussed in the actual manuscript. This control measurement is vital to the conclusions of the authors and needs to be included.
  3. Thanks to the authors for providing SEM images. However, images B and D repeat the same problem mentioned previously: that the scale bars are different (and remain different in the manuscript, see Figs 2A-B). I genuinely cannot differentiate any differences between A and C at the scale provided in a manner that supports the cartoon of Figure 9. In total, I find it difficult to rationalize the SEM images presented by the authors in Figures 2A-B, as well as in their response, with the data in Figure 4. The latter Figure indicates a great degree of size matching between NCM and GN13 relative to GN28, and it is easy to see the NCM distribution of Figure 4 in the SEM provided as Figure 1B. It is very hard to say the same for the graphene distributions when only one flake is imaged in Figure 2A and it looks to be of comparable size to those in Figure 2B. And again- such data were collected only for my review and not included in the manuscript.
  4. "The GN-13 and GN-28 are produced with the same material"- what does this mean? How does the same reagent produce two quantitatively different lateral sizes of graphene? I'm not sure how to read the ICP data here.

Author Response

Comment 1:

Regarding the authors response: The authors satisfactorily addressed the electrode preparation procedure. The data procured by the authors appears to show their graphene indeed improving the electrode relative to super P carbon. However, such a Figure is provided only for my review and is not included or discussed in the actual manuscript. This control measurement is vital to the conclusions of the authors and needs to be included.

Response:

Thank you for reviewer’s suggestion. We have added the control measurement by using only super P carbon as conductive additives (5 wt%) in NCM electrode. As shown in the following figure. The corresponding discussion have also included in the revised manuscript.

For comparison, we also introduced 5 wt% Super P as a conductive additive in NCM electrode for the control measurement. The Super-P electrode shows reversible capacity of 168.4, 151.4, 136.6 and 37.2 mAh/g at 0.1, 0.5 ,1 and 2C, respectively. Compare to GN-13/Super-P and GN-28/Super-P, we can found that bi-conductive additives gave much better rate capability than Super P. For example, at 2C, the capacity of GN-13 and Super P were 114.2 mAh/g and 37.2 mAh/g. The enhancement could be 3.06 times by intruding GN-13/Super P as composite conductive additives. The reason for the dramatically enhancement might be explained that GN-13/Super P provided a point-to-plane structure provide shorter Li-ion transfer path. The structure of Super-P electrode is only point-to-point structure that the C-rate performance was worse at high rate. The results show that the high rate performance of the electrode become worse without the addition of graphene.

Comment 2:

Thanks to the authors for providing SEM images. However, images B and D repeat the same problem mentioned previously: that the scale bars are different (and remain different in the manuscript, see Figs 2A-B). I genuinely cannot differentiate any differences between A and C at the scale provided in a manner that supports the cartoon of Figure 9. In total, I find it difficult to rationalize the SEM images presented by the authors in Figures 2A-B, as well as in their response, with the data in Figure 4. The latter Figure indicates a great degree of size matching between NCM and GN13 relative to GN28, and it is easy to see the NCM distribution of Figure 4 in the SEM provided as Figure 1B. It is very hard to say the same for the graphene distributions when only one flake is imaged in Figure 2A and it looks to be of comparable size to those in Figure 2B. And again- such data were collected only for my review and not included in the manuscript.

Response:

Thank you for reviewer’s comments. We completely agree with reviewer’s suggestions. We have slightly changed the magnitudes of SEM and TEM images in the revised Figure 2(a) to (d). Please see the following revised Figure.

As for reviewer’s comment: “It is very hard to say the same for the graphene distributions when only one flake is imaged in Figure 2A and it looks to be of comparable size to those in Figure 2B.” We mush to say NCM cathode is a zero-dimensional materials (sphere-like structure). Thus, it is very easy to demonstrate their particle size distribution data from SEM (Fig. 1(b)) and laser scattering measurement (Fig. 1(c)). However, graphene is a two-dimensional material with very large aspect ratio. For few-layer graphene, said <10 layers, the structure of graphene is a folded structure. It is difficult to obtain flat graphene without folding. We hope reviewer’s could understand and satisfy our response.

In order to let reader more understand the distribution of NCM, GN-13 and GN-28. In Fig. 9, we propose the SEM images of fresh electrodes (without charge and discharge) of GN-13 electrode (Fig. 9(a)~(b)) and GN-28 electrode (Fig. 9(c)~(d)). The red ones are NCM. The blue ones and green ones are GN-13 and GN-28, respectively. Because all these electrodes are pressed by roller. Thus, it is a little difficult to identify which one is cathode material (NCM) and which one is conductive additives (GN-13, GN-28 or Super P). Thus, we do our best to label them. It may be a very good reference to demonstrate the distribution of NCM cathode and GN-based conductive additives.

Comment 3:

"The GN-13 and GN-28 are produced with the same material"- what does this mean? How does the same reagent produce two quantitatively different lateral sizes of graphene? I'm not sure how to read the ICP data here.

Response:

GN-13 and GN-28 are produced by a liquid exfoliation process. The corresponding experimental detail reported and published in Scientific Reports, 8, 9766, 2018 [1]. The precursor we used is nature graphite (NG). By changing chamber-pressure and cycling tests, we can obtain few layer graphene with different lateral size. (Please see the following results)

In order to guarantee quality and purity of as-synthesized few layer graphene. We measure the composition of our starting materials, graphite, by ICP. The following table summarized the possible impurity and their concentration in graphite. As you can see, the concentrations of these impurity were less than 1054 ppm. We use the same “mother” to produce few layer graphene by a cavitation process (liquid exfoliation). Thus, we believe the purity of GN-13 and GN-28 should be the same. The electrochemical comparison of rate capability by using GN-13 and GN-28 in NCM electrode is reliable.

Reference

[1] Pin-Chun Lin, Jhao-Yi Wu and Wei-Ren Liu, “Green and facile synthesis of few-layer graphene via liquid exfoliation process for Lithium-ion batteries,” Scientific Reports, 8, 9766 (2018).

Round 3

Reviewer 3 Report

Thanks to the authors for their changes.

My final comment on this paper: the reference regarding the graphene synthesis is surprising considering that the experimental section had previously stated the graphene was purchased commercially. Having looked at the Sci Rep 2018 paper, there is no discussion in that manuscript on how to change the lateral graphene dimension, so a simple citation to ref. 47 is insufficient reporting of experimental details. The authors should list the chamber pressure and cycles used in the experimental section. 

Author Response

Review 3

Comment 1:

My final comment on this paper: the reference regarding the graphene synthesis is surprising considering that the experimental section had previously stated the graphene was purchased commercially. Having looked at the Sci Rep 2018 paper, there is no discussion in that manuscript on how to change the lateral graphene dimension, so a simple citation to ref. 47 is insufficient reporting of experimental details. The authors should list the chamber pressure and cycles used in the experimental section. 

Response:

Thank you for reviewer’s kindly reminding. The detail description how to synthesize GN-13 and GN-28 samples in terms of chamber pressure and cycles have been added in the revised experimental section.

For comparison, two different lateral sizes of graphene nanosheets (denoted as GN-13 and GN-28) are synthesized by using the cavitation process via fixing the chamber pressure at 2000 bar and changing the cycling times by 3 times or 12 times to obtain GN-13 (smaller lateral size) and GN-28 (bigger lateral size), respectively [47].

We are sincerely grateful for your time and consideration.

Respectfully yours,

Wei-Ren Liu,

Professor,

Department of Chemical Engineering,

Chung Yuan Christian University
